# Exploring the Hydrogen-Induced Amorphization and Hydrogen Storage Reversibility of Y(Sc)_0.95_Ni_2_ Laves Phase Compounds

**DOI:** 10.3390/ma14020276

**Published:** 2021-01-07

**Authors:** Shiqian Zhao, Hui Wang, Jiangwen Liu

**Affiliations:** School of Materials Science and Engineering, Guangdong Provincial Key Laboratory of Advanced Energy Storage Materials, South China University of Technology, Guangzhou 510641, China; 201710103269@mail.scut.edu.cn (S.Z.); mejwliu@scut.edu.cn (J.L.)

**Keywords:** hydrogen storage alloys, hydrogen-induced amorphization, Sc substitution, reversibility

## Abstract

Rare-earth-based AB_2_-type compounds with Laves phase structure are readily subject to hydrogen-induced amorphization and disproportionation upon hydrogenation. In this work, we conducted the Sc alloying on Y_0.95_Ni_2_ to improve its hydrogen storage properties. The results show that the amorphization degree of Y_0.95_Ni_2_ deepens with the increasing hydrogenation time, pressure, and temperature. The Y(Sc)_0.95_Ni_2_ ternary compounds show a significant improvement in reversibility and dehydriding thermodynamics due to the reduced atomic radius ratio R_A_/R_B_ and cell volume. Hydrogen-induced amorphization is fully eliminated in the Y_0.25_Sc_0.7_Ni_2_. The Y_0.25_Sc_0.7_Ni_2_ delivers a reversible hydrogen storage capacity of 0.94 wt.% and the dissociation pressure of 0.095 MPa at the minimum dehydrogenation temperature of 100 °C.

## 1. Introduction

Numerous intermetallic compounds in the formula of AB_n_, in which A is the strong hydride-forming elements and B is weak hydride-forming elements, have been considered to be a potential hydrogen storage medium for either gaseous or electrochemical applications [1,2,3,4,5,6]. Among them, AB_2_-type intermetallic compounds possess higher theoretical specific capacity than the most widely used AB_5_-type rare-earth compounds. Zr-based and Ti-based compounds with cubic MgCu_2_-type (C15) or hexagonal MgZn_2_-type (C14) structure could store more than 2.0 wt.% hydrogen [7,8,9]. However, the rare-earth-based AB_2_-type compounds have rarely been considered for reversible hydrogen storage because of the hydrogen-induced amorphization (HIA) and phase disproportionation (HID) upon hydrogenation [10,11,12,13,14,15,16,17]. For example, the YFe_2_ with cubic C15 structure would experience four steps of structural change in the hydrogenation: the formation of crystalline hydride YFe_2_H_x_, the transformation of crystalline YFe_2_H_x_ to non-crystalline YFe_2_H_x_, precipitation of YH_3_ from non-crystalline hydride, and the decomposition into YH_3_ and α-Fe [18].

The HIA and HID mechanisms of several representative rare-earth-based AB_2_-type compounds have been comprehensively investigated in early studies by Aoki et al. [19,20]. The hydrogenation pressure, time, and temperature dependence has been demonstrated. For example, the structural change of DyNi_2_ during hydrogen absorption is dependent on the pressure. The decomposition phases DyNi_5_ and DyH_3_ were finally obtained above 2.0 MPa hydrogen [21]. The occurrence of HIA and HID means the structural instability of AB_2_-type compounds, especially as hydrogen atoms enter the tetrahedral interstices and cause the lattice expansion. The structural stability of rare-earth-based AB_2_-type compounds is closely related to the atomic radius ratio R_A_/R_B_ of the constituting A and B elements. The ideal R_A_/R_B_ value for the cubic Laves phase structure is 1.225 [22], and the larger the R_A_/R_B_ deviation from the value of 1.225, the more unstable Laves phase structure. Aoki et al. proposed a critical R_A_/R_B_ value of 1.37 as the judgment factor of HIA by summarizing a large variety of AB_2_-type hydrogen storage compounds [20]. Since rare-earth elements possess a relatively large atomic radius, most rare-earth-based AB_2_-type compounds have a R_A_/R_B_ value exceeding 1.37 and, thus, are readily subject to HIA.

Rare-earth-based compounds form the largest subset of the AB_2_-type Laves phase compounds [23], and their potential for hydrogen storage is worthy of more exploration to overcome the HIA problem by compositional and structural modification. Our previous studies show that the B-side substitution with Al and the A-side alloying with Zr could effectively inhibit the HIA and achieve reversible hydrogen storage [24,25,26,27]. The YFe_1.7_Al_0.3_ delivers a reversible hydrogen storage capacity of 1.38 wt.%, but the complete hydrogen desorption must be conducted in a dynamic vacuum at 200 °C [24]. The Zr alloying could realize the reversible hydrogen absorption with the elevated dissociation pressure of YFe_2_ because of the reduced cell volume [25]. Zhang et al. [28] reported that the pseudo-binary Sm_1.25_Mg_0.75_Ni_4_ compound could reversibly absorb and desorb ca. 0.95 wt.% hydrogen without the occurrence of HIA at 25 °C.

Regarding the Y_0.95_Ni_2_ compound with a C15 superlattice structure, the crystalline hydride Y_0.95_Ni_2_H_2.6_ with a capacity of 1.27 wt.% has been reported, but further increasing the hydrogen capacity would result in amorphization [29]. Zhang et al. reported that the Y_0.95_Ni_2_ experienced gradual amorphization during absorption/desorption cycling under gaseous hydrogen at ambient temperature [30]. In this work, we conducted the Sc alloying on Y_0.95_Ni_2_ to suppress the HIA because the Sc element has a relatively smaller atomic radius (1.66 Å) than Y (1.78 Å), and, thus, the R_A_/R_B_ value could be decreased. The effect of Sc content on the microstructure and hydrogen storage properties of Y–Sc–Ni compounds have been investigated.

## 2. Materials and Methods

Commercial Y, Sc, and Ni ingots with a purity of 99.9% were used as the starting materials. Y_0.95-x_Sc_x_Ni_2_ (x = 0, 0.1, 0.3, 0.5, 0.6, 0.7) alloys were prepared by arc melting under a high purity Argon atmosphere with Y, Sc, and Ni bulks in proper proportion. All samples were remelted four to five times and annealed at 950 °C for 4 days to obtain homogenous composition and structure. The alloys were crushed into powder and sieved by a 200-mesh (~74 um) sieve in the glove box for further measurement.

The composition and microstructure were observed using a scanning electron microscope (SEM, Zeiss Supra 40/VP, Carl Zeiss AG, Jena, Germany) equipped with an energy-dispersive spectrometer (EDS). X-ray diffraction (XRD) measurements were conducted using a PANalytical Empyrean with Cu Ka radiation (λ = 1.5406 Å, PANalytical B.V., Almelo, The Netherlands). Rietveld refinements were conducted by using the Highscore Plus 4.5 software (PANalytical B.V.).

To investigate hydrogen-induced amorphization, the alloy powder was loaded into stainless steel reactors in an Argon-filled glove box. Then, the alloy powder was subjected to a hydrogenation reaction under different hydrogenation conditions: 3 MPa H_2_-100 °C-2 h, 3 MPa H_2_-100 °C-24 h, 3 MPa H_2_-200 °C-2 h, and 5 MPa H_2_-100 °C-24 h. The hydrogen desorption and crystallization processes of hydrogenated samples were monitored by differential scanning calorimetry (DSC, Setaram SENSYS Evolution, SETARAM Instrumentation, Lyon, France) at a heating rate of 5 °C/min.

The measurement of hydrogen storage properties was conducted on an automatic Sieverts-type apparatus (AMC HP2000, Advanced Materials Corporation, Pittsburgh, America). About a 1.3 g sample was put into the sample holder, the hydriding, and dehydriding pressure-composition isotherms (PCI) of Y_0.25_Sc_0.7_Ni_2_ compound were measured at different temperature.

## 3. Results

### 3.1. Structure of Y(Sc)_0.95_Ni_2_ Compounds

Figure 1 shows the XRD patterns of as-annealed Y_0.95-x_Sc_x_Ni_2_ (x = 0, 0.1, 0.3, 0.5, 0.6, 0.7) compounds. The structural parameters of all compounds are refined by the Highscore Plus 4.5 software and summarized in Table 1. For the compounds with the Sc content of 0, 0.1, and 0.3, in addition to the strong Bragg peaks of cubic C15 Laves phase (MgCu_2_-type structure, space group *Fd*$\overset{—}{3}$*m*), some extra weak Bragg peaks indicate the formation of the C15 superstructure (C15_s_) with the space group of *F*$\overset{—}{4}$*3m*. The C15_s_ structure of Y_0.95_Ni_2_ has a doubled lattice constant of 14.35 Å compared with the C15 structure. The structural refinement by Latroche et al. shows that most atoms in the C15_s_ structure are shifted from the ideal atomic positions expected from the C15 structure [31]. It is noted that the Y vacancies in the 4a site could stabilize the Laves phase structure of nonstoichiometric Y_0.95_Ni_2_ with a large R_A_/R_B_ value of 1.424 [29]. Since the Sc content is larger than 0.3, the weak peaks of the superstructure disappear, implying the transformation from the C15_s_ superstructure to the C15 structure. This transformation should be related to the decrease of the R_Y_/R_Ni_ value due to Sc substitution, which results in the decrease of the number of ordered Y vacancies [31]. The Rietveld refinement results indicate that the 4a site occupancy of the Y vacancy for Y_0.85_Sc_0.1_Ni_2_ is 0.421, which decreases to almost zero for the Y_0.65_Sc_0.3_Ni_2_. The Y_0.25_Sc_0.7_Ni_2_ has an R_A_/R_B_ value of 1.353, which is much lower than that of the Y_0.95_Ni_2_. Therefore, the ScNi_2_ phase with the C15 structure is identified for the Y_0.25_Sc_0.7_Ni_2_ [32]. EDS analysis (Figure 2) shows that the representative compound Y_0.__25_Sc_0.__7_Ni_2_ has a homogeneous microstructure and distribution of Y and Sc elements. Further, the EDS compositions of each Y–Sc–Ni compound is highly consistent with the designed ones, as shown in Table 1.

### 3.2. Hydrogen-Induced Amorphization of Y_0.95_Ni_2_

The effect of hydrogenation temperature, time, and pressure on the HIA of Y_0.95_Ni_2_ is investigated. Figure 3 compares the structures of Y_0.95_Ni_2_ hydrogenated under different conditions with the as-annealed Y_0.95_Ni_2_. After hydrogenation at 100 °C-3 MPa-2 h, Figure 3b shows that all Bragg peaks shift toward the low angle direction, indicating the formation of crystalline hydride of Y_0.95_Ni_2_. The interstitial hydride also adopts the same C15_s_ structure. As shown in Figure 3c,d there exists a broad scattering peak at 30° to 45° in addition to the diffractions of crystalline hydride, indicating the formation of amorphous phase at an elevated temperature of 200 °C or prolonged time of 24 h. It is noted that the present Y_2_O_3_ phase may be related to the partial oxidization of Y during the arc-melting or hydrogenation. When hydrogenated at 100 °C-5 MPa-24 h, the hydrogenated product consists of the major amorphous phase, indicating full transformation from crystalline hydride to non-crystalline hydride under the elevated hydrogenation pressure. Hence, it is assumed that the increased hydrogen storage capacity due to the increased hydrogenation temperature, pressure, and time results in the destruction of the Laves phase structure of Y_0.95_Ni_2_ hydride, which is in line with other rare-earth RENi_2_ compounds [14].

### 3.3. Hydrogen-Induced Amorphization of Y–Sc–Ni Compounds

Figure 4 shows the XRD patterns of Y_0.95−*x*_Sc*_x_*Ni_2_ (*x* = 0.1, 0.3, 0.5, 0.6, 0.7) compounds after hydrogenation at 100 °C-5 MPa-24 h. Like the Y_0.95_Ni*_2_*, the full HIA occurs for the Y_0.85_Sc_0.1_Ni_2_. With the increasing Y content of 0.3–0.6, the Bragg peaks of crystalline hydride with a C15 structure become intensifying, indicating that the HIA is partially suppressed due to the Sc alloying. The absence of the broadening scattering peak and the presence of C15 diffractions indicate that the HIA is fully eliminated in the hydrogenation of Y_0.25_Sc_0.7_Ni_2_. As shown in Table 1, the R_A_/R_B_ value of Y–Sc–Ni compounds decreases with increasing Sc content, being 1.353 for the Y_0.25_Sc_0.7_Ni_2_. This result implies that the Sc alloying could improve the structural stability of Y_0.95_Ni_2_, which should be the structural reason for eliminating HIA.

Figure 5a further shows the DSC profiles of Y–Sc–Ni compounds hydrogenated under 100 °C-5 MPa-24 h. The XRD patterns of DSC samples at specific temperatures are shown in Figure 5b. As shown in Figure 5a, a rather broad endothermic peak is followed by a sharp exothermic peak, which is attributed to the dehydrogenation of non-crystalline hydride and the crystallization of non-crystalline hydride, respectively [33]. This assumption is further confirmed by XRD. At the T1 temperature of the DSC curve of the Y_0.95_Ni_2_, no crystalline phase exists in addition to the Y_2_O_3_. At the T2 temperature, the presence of YH_2_ and YNi_3_ phases indicates the crystallization of non-crystalline hydride and subsequent decomposition. Similar DSC results are also reported for the GdNi_2_ [34], SmNi_2_ [35], and YNi_2_ [36] compounds. With the increasing Sc content, the broad endothermic peak contains a sharp part with an increasing shaded area. The broad endothermic peak below the broken line still corresponds to the hydrogen desorption from noncrystalline hydride, while the sharp endothermic peak with a shaded area is attributed to hydrogen release from the crystalline hydride [33]. In addition, the exothermic peak area and the peak temperature decrease with increasing Sc content. XRD patterns of DSC samples at T3, T4, T5, and T6 also show the weak peaks of YH_2_ and YNi_3_ phases, indicating that the amount of non-crystalline hydride decreases with increasing Sc content. For the hydrogenated Y_0.25_Sc_0.7_Ni_2_, the only sizeable endothermic peak in the DSC curve further verifies the elimination of HIA.

### 3.4. Hydrogen Storage Properties of Y_0.25_Sc_0.7_Ni_2_

Figure 6a displays the hydriding and dehydriding PCI curves of Y_0.25_Sc_0.7_Ni_2_ at different temperatures. There exists a clear plateau region in the hydriding curve, but the dehydriding curve starts to slope. At the midpoint of the PCI curve, the hydriding and dehydriding equilibrium pressures are determined to calculate the enthalpy change *ΔH* and entropy change *ΔS*, and the fitted van’t Hoff plots are displayed in Figure 6b. The derived dehydriding *ΔH* and *ΔS* for the Y_0.25_Sc_0.7_Ni_2_ are 35.59 kJ/mol·H_2_ and 93.83 J/K/mol·H_2_, respectively. In addition, the Y_0.25_Sc_0.7_Ni_2_ exhibits a reversible hydrogen storage capacity of 0.94 wt.% at 100 °C, corresponding to the hydride formula Y_0.25_Sc_0.7_Ni_2_H_1.92_. However, a small amount of hydrogen could not be released at 80 °C, which should be related to the low dissociation pressure. It is strange that, in the present work, the hydride of Y_0.25_Sc_0.7_Ni_2_ has a lower capacity than the Y_0.9__5_Ni_2_D_2.6_ in Reference [29], which may be related to the increased equilibrium hydrogen pressure at a higher testing temperature of 100 °C. It is noted that the Y_0.95_Ni_2_D_2.6_ was determined at ambient temperature and deuterium pressure. As shown in the hydriding kinetic curves at 40 °C in Appendix A, the hydrogen absorption capacity increases and then decreases with Sc content. The Y_0.65_Sc_0.3_Ni_2_ delivers a maximum hydrogen absorption capacity of 1.53 wt. %, corresponding to the hydride formula Y_0.65_Sc_0.3_Ni_2_H_2.93_. It is noted that the minimum dehydrogenation temperature of the Y_0.25_Sc_0.7_Ni_2_ is much lower than that of Y(Zr)Fe_2_ compounds [24,25,26,27]. Therefore, in addition to the improvement in the reversibility due to the reduced R_A_/R_B_, the Sc alloying also shows a significant promoting effect in the hydrogen sorption equilibrium pressure because of the reduced lattice constant.

## 4. Conclusions

The gaseous hydrogen storage reversibility of Y(Sc)_0.95_Ni_2_ compounds with different Sc contents has been investigated. The hydrogen-induced amorphization of Y_0.95_Ni_2_ is dependent on the hydrogenation pressure, time, and temperature. The Sc alloying could improve structural stability and suppress the hydrogen-induced amorphization. Fully reversible hydrogen storage has been achieved for the Y_0.25_Sc_0.7_Ni_2_, with a reversible capacity of 0.94 wt.% and dissociation pressure of 0.095 MPa at the minimum dehydrogenation temperature of 100 °C. Our work demonstrates the feasibility of rare-earth-based AB_2_-type compounds for reversible hydrogen storage applications.

## Figures and Tables

**Figure 1 materials-14-00276-f001:**
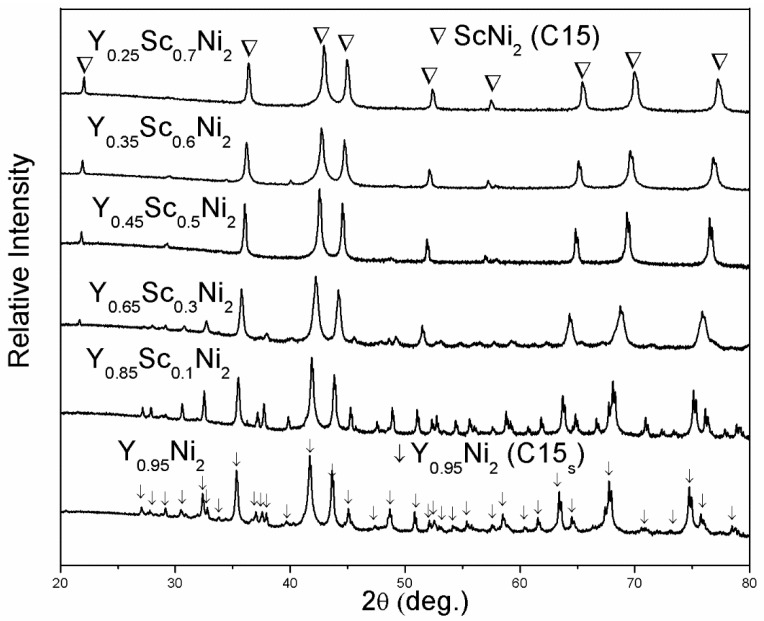
XRD patterns of as-annealed Y-Sc-Ni compounds.

**Figure 2 materials-14-00276-f002:**
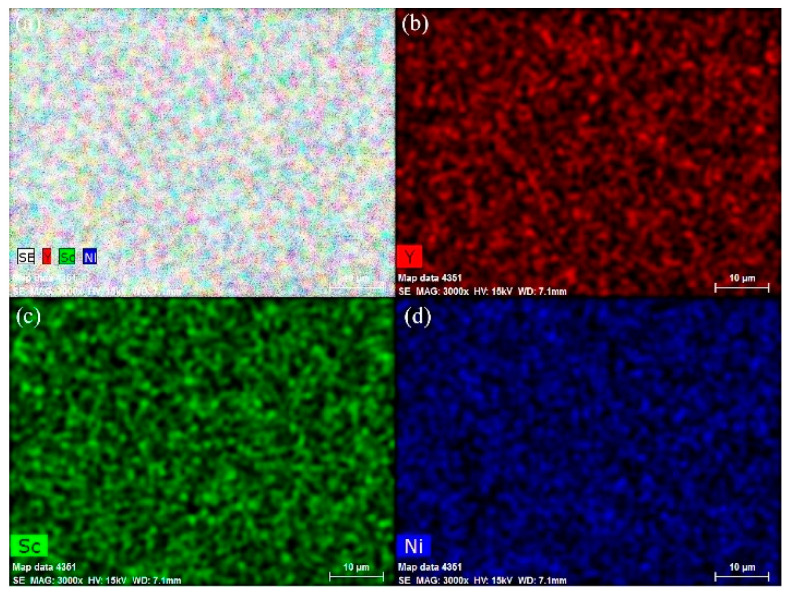
EDS elemental mapping (**a**–**d**) of as-annealed Y_0.__25_Sc_0.__7_Ni_2_.

**Figure 3 materials-14-00276-f003:**
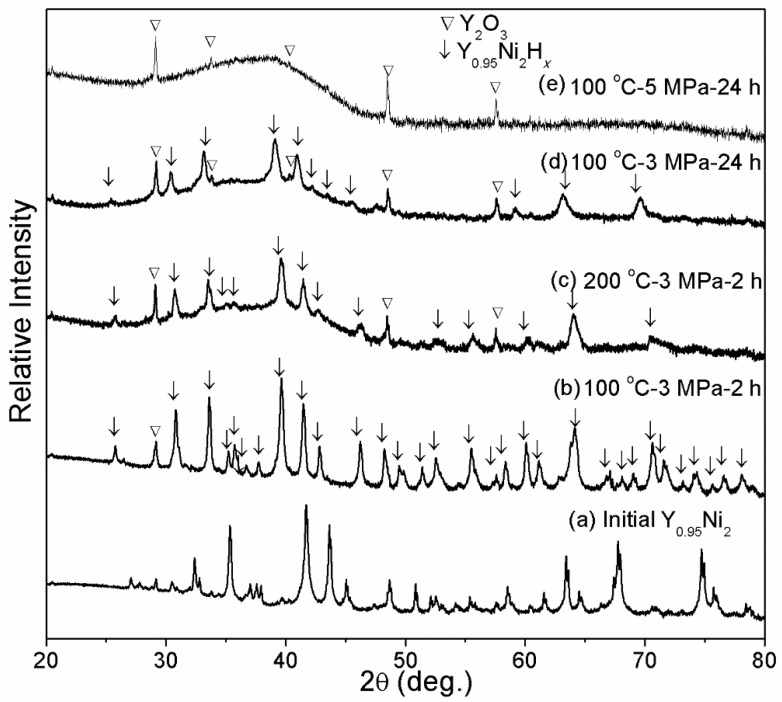
XRD patterns of Y_0.95_Ni_2_ after hydrogenation under different temperatures, hydrogen pressure, and times. (**a**) Initial Y_0.95_Ni_2_; (**b**) 100 °C-3 MPa-2 h; (**c**) 100 °C-3 MPa-2 h; (**d**) 100 °C-3 MPa-24 h; (**e**) 100 °C-5 MPa-24 h.

**Figure 4 materials-14-00276-f004:**
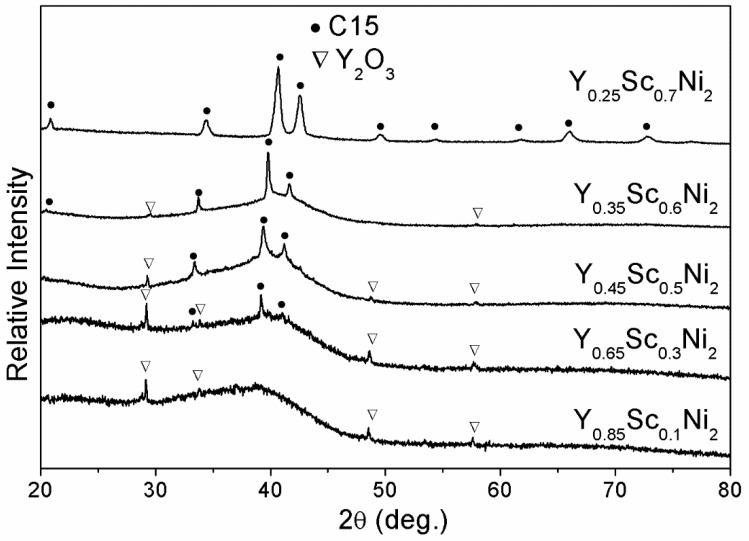
XRD patterns of Y–Sc–Ni compounds after hydrogenation at 100 °C-5 MPa-24 h.

**Figure 5 materials-14-00276-f005:**
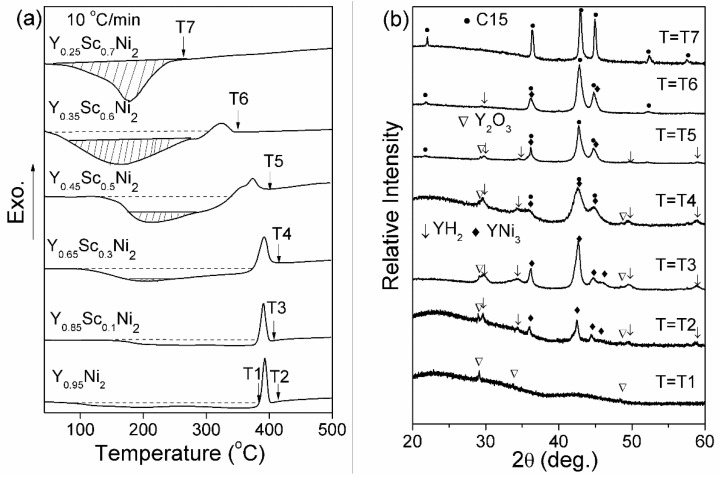
Differential scanning calorimetry (DSC) profiles of Y–Sc–Ni compounds after hydrogenation at 100 °C-5 MPa-24 h (**a**) and the corresponding XRD patterns (**b**).

**Figure 6 materials-14-00276-f006:**
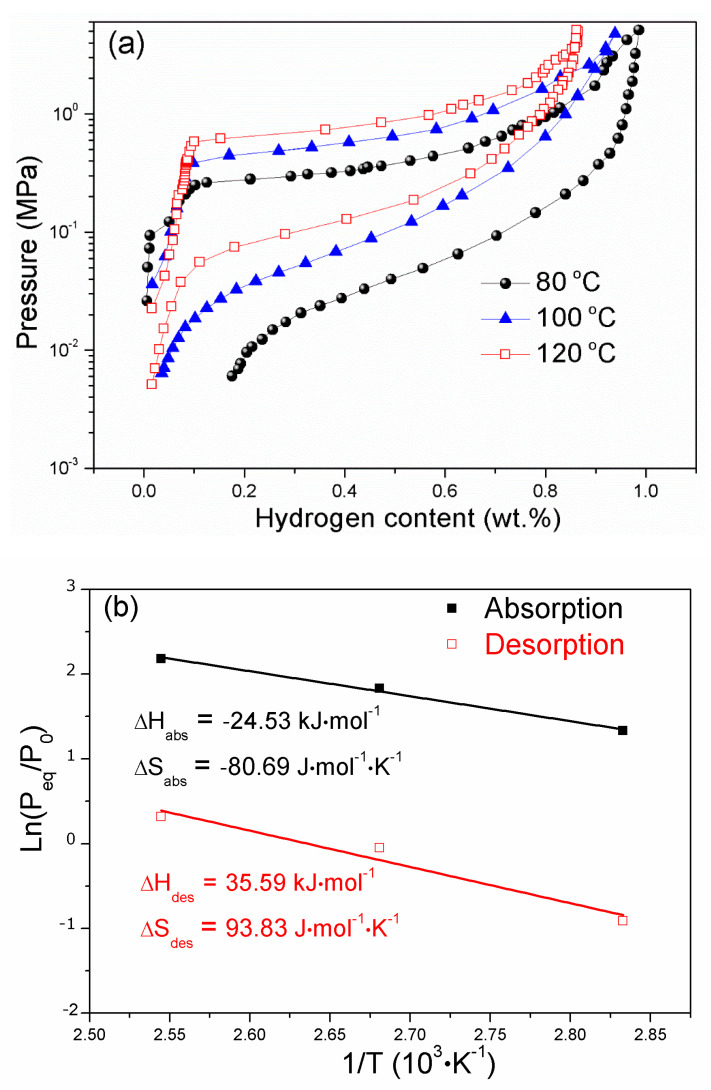
Hydriding and dehydriding pressure-composition isotherms (PCI) curves (**a**) and van’t Hoff plots (**b**) for Y_0.25_Sc_0.7_Ni_2_.

**Table 1 materials-14-00276-t001:** Structural parameters of Y_0.95−*x*_Sc*_x_*Ni_2_ compounds.

Nominal Composition	EDS Composition	Space Group	Crystal Structure	Lattice Constant, a/Å	* R_A_/R_B_	HIA
Y_0.95_Ni_2.00_	Y_0.96(1)_Ni_2.00(3)_	*F* $\overset{—}{4}$ *3m*	C15s	14.35	1.424	Yes
Y_0.85_Sc_0.10_Ni_2.00_	Y_0.84(1)_Sc_0.11(1)_Ni_2.00(3)_	*F* $\overset{—}{4}$ *3m*	C15s	14.29	1.414	Yes
Y_0.65_Sc_0.30_Ni_2.00_	Y_0.66(1)_Sc_0.30(1)_Ni_2.00(3)_	*F* $\overset{—}{4}$ *3m*	C15s	14.15	1.394	partial
Y_0.45_Sc_0.50_Ni_2.00_	Y_0.46(1)_Sc_0.50(1)_Ni_2.00(3)_	*Fd* $\overset{—}{3}$ *m*	C15	7.03	1.373	partial
Y_0.35_Sc_0.60_Ni_2.00_	Y_0.36(1)_Sc_0.60(1)_Ni_2.00(3)_	*Fd* $\overset{—}{3}$ *m*	C15	7.01	1.363	partial
Y_0.25_Sc_0.70_Ni_2.00_	Y_0.26(1)_Sc_0.70(1)_Ni_2.00(3)_	*Fd* $\overset{—}{3}$ *m*	C15	6.98	1.353	No

* R_A_ = $R_{Y}\times \frac{0.95-x}{0.95}+R_{\mathrm{Sc}}\times \frac{x}{0.95}$, R_B_ = R_Ni_ = 1.25 Å, R_Y_ = 1.78 Å, R_Sc_ = 1.66 Å.

## Data Availability

Not applicable.

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
