# Peer review of "Exploring the Hydrogen-Induced Amorphization and Hydrogen Storage Reversibility of Y(Sc)0.95Ni2 Laves Phase Compounds"

_materials, 2021, doi:10.3390/ma14020276_

Round 1

Reviewer 1 Report

This work reports on the structural and hydrogenation properties of (Y,Sc)0.95Ni2 pseudobinary compounds. The aim of the partial Sc for Y substitution is to decrease the RA/RB ratio of the Laves phase compounds to reduce/eliminate the hydrogen induced amorphisation observed for Y0.95Ni2 compound and favor a reversible hydrogen absorption. This purpose is achieved for a large Sc substitution rate with the Y0.25Sc0.7Ni2 composition.

The work is in general clearly presented and contains original results which deserves publication in Materials, but several points need to be clarified and both minor and major revisions are required.

On the form: the figures in the formula should be as subscript for page 1 to 3 ex: MgCu2 should be MgCu2 .

page 1, line 41: the Aoki reference number is missing.

page 2, lines 48-50: “The greater deviation … in the hydrogenation”  this sentence is not very clear and should be rewritten. For example it is not the deviation which is unstable

line 49 Correct Lave by Laves

page 2, lines 84-88 check the writting of °

page 3 line 104 replace angel by angle

line 106: the sentence “As the Sc content …are still present”

Is there only 2 superstructure peaks to define that ScNi2 crystallizse in the superstructure, is it enough?

A discussion about the decrease of the superstructure peak intensity observed in Figure 1 should be added. It would be also very interesting in this case to have complete fullprof refinement to estimate not only the cell parameter variation but also the quantity of ordered vacancies as the Sc rate increases. This evolution could be also related to the decrease of the average RA value (RA/RB ratio) as it was observed in the full RNi2 study versus R radius.

Table 1: what is the precision of the EDS composition? The composition are reported with 2 decimal numbers, but this method is not as accurate as Microprobe analysis. Add some precision

From Figure 1: the rate of ordered vacancies decreases, but the composition analysis by EDS indicates that there is always 5 % of vacancies on the A site, whatever the Sc content. Does it mean that there are still disordered vacancies in Y0.25Sc0.7Ni2?

Table 1: the symbols used for HIA,are not so clear. It can be better replaced by Yes or No.

Line 119: RM should be replaced by RSc.

line 165 “the weakening peaks“ : “the weak peaks” or “the weakening of the peaks”?

line 166 “is decreased” decreases or is decreasing?

In the description of the hydrogenation properties, the quantity of absorbed hydrogen as a function of temperature and pressure is missing. As you did PCI measurements and use a Sievert apparatus you should be able to determine the H content for the hydrides presented in Figures 3, 4 and 5.

Y0.25Sc0.7Ni2H1.92 contains less H/f.u than Y0.94Ni2D2.6  [26]. It would be interesting at least to determine the H content in the reversible crystalline hydrides as a function of Sc content to see if it is reduced versus Sc content and to check if there is a limit of H concentration before the amorphisation which occurs at higher pressure and H content.  As explained later the advantage of Sc is to reduce the cell parameter and therefore increase the equilibrium pressure which allow to desorb hydrogen than for Y0.94Ni2D2.6.

Reviewer 2 Report

Hydrogen storage capacity of the studied materials are nowhere close to DOE or any other hydrogen storage targets for mobile applications. Authors should discuss this shortcoming of this material and if possible, offer alternative applications areas.

Any previous studies on Sc alloying in the literature, not much discussed in intro

Any more studies on Ra/Rb ratio in the literature? Is Ra/Rb the most important factor for HID/HIA, discussing more would be helpful to the readers.

 Fig. 2: Each sub figure window should be labeled as "a", "b", "c" etc. What is SF stands for on the image to the left of Y. Also, gray image is not useful, consider removing so elemental distribution images would look bigger.

Reversibility term is not very clear or misused in the text. Usually reversibility is reserved for multiple cycles and their effect on hydrogen storage capacity and it is an important performance metric. Authors discussed only single cycle performance, but authors use reversibility for just a single cycle. This point should be clarified in text.

I recommend including additional cycle data if available so we can better understand how this materials perform over multiple cycles. Is performance degrading over time?

Do you have XRD images after one cycle? Is structure preserved?

Editorial comments

All authors look like from the same institution, if that’s the case why did you include three identical affiliations for each author separately?

There are multiple typos, such as:

  • Laves on page 2/ line 49 missing s
  • Page 3/line 104, supposed to read "angle" not "angel"
  • Page /line 108, there is an extra "a" before "of"
  • Some of the chemical formulas have issues with subscripts/superscripts - verify
  • All degree symbols looks like "@" symbol in text - check
  •  

Reviewer 3 Report

This work investigates hydrogen sorption properties of ScxY0.95−xNi2 intermetallics in an attempt to improve the hydrogen storage properties of Y0.95Ni2, as well as to reduce and even eliminate hydrogen-induced amorphization (HIA).  

Several comments of technical character follow:

- All the authors have the same affiliation. It is then redundant to write it three times after their names; once is enough.

- row 29: It is suggested to write “B is a weak hydride-forming element” instead of “B is non hydride-forming elements”. For example, Ni forms a hydride at very high pressures with enthalpy and entropy of formation of about −8.8 (kJ/mole H2) and −106.3 (J/(K×mole H2), respectively.

- Row 46: It should be written “tetrahedral interstices” instead of “tetragonal interstices”.

- row 49: It should be written “Laves” instead of “Lave”. The whole sentence should be formulated more clearly from a grammatical point of view.

- row 47: It should be written “atomic radius ratio” instead of “atomic ratio”.

- row 119: rA and rB should be defined somewhere in the text as are RA and RB.    

- row 181: It it is suggested to write “than in the” instead of “than the”.

- The authors have extensively covered the relevant literature associated with HIA. Together with this, it may be turned to their attention that the loss of ordered XRD patterns after hydrogenation (demonstrating HIA) had been reported in some earlier papers. For example, a disappearance of the XRD crystalline pattern upon hydrogenation has been observed in LaNi2 (H. Oesterreicher, J. Clinton, H. Bittner, Hydrides of La-Ni compounds, Mater. Res. Bull. 11(1976)1241-1248), GdNi2 (S.K. Malik , W.E. Wallace, Hydrogen absorption and its effect on structural and magnetic behavior of GdNi2, Solid State Comm. 24(1977)283-285), in GdB2, B=Mn, Fe, Co, Ni (I. Jacob, D. Shaltiel, Hydrogen sorption properties of some AB2 Laves phase compounds, J. Less-Comm. Met. 65(1979)117-128).  It is though not necessarily to include these references in their paper.

Round 2

Reviewer 1 Report

The authors have made  appropriate changes to improve the manuscript. I agree for manuscript publication after the following changes:

I propose that the Rietveld results given in the answer to author page 4, are added in the manuscript (as text to avoid a new table) as it is an important information.

The figure of the hydriding kinetic at 40 °C is also interesting and can be added in the manuscript or as suplementary material.
